# Very Low Protein Diet for Patients with Chronic Kidney Disease: Recent Insights

**DOI:** 10.3390/jcm8050718

**Published:** 2019-05-20

**Authors:** Lucia Di Micco, Luca Di Lullo, Antonio Bellasi, Biagio R. Di Iorio

**Affiliations:** 1Nefrology and Dialysis, AORN “San Giuseppe Moscati, 83100 Avellino AV, Italy; luciadimicco@gmail.com; 2Department of Nephrology and Dialysis, Parodi-Delfino Hospital, 00034 Colleferro, Rome, Italy; dilulloluca69@gmail.com; 3Department of Research, Innovation and Brand Reputation, ASST Papa Giovanni XXIII, 24127 Bergamo BG, Italy; abellasi@asst-pg23.it; 4Nephrology and Dialysis, AORN “Antonio Cardarelli”, 80131 Napoli, Italy

**Keywords:** chronic kidney disease, nutritional therapy, urea, phosphorus, metabolic acidosis, vascular calcification, proteinuria, gut, microbioma, cardiovascular risk, very low protein diet

## Abstract

Use of nutritional therapy (NT) in chronic kidney disease (CKD) patients is still debated among nephrologists, but it represents a fundamental point in the conservative treatment of CKD. It has been used for years and it has new goals today, such as (1) the reduction of edema, diuretics, and blood pressure values with a low sodium-content diet; (2) the dose reduction of phosphate levels and phosphate binders; (3) the administration of bicarbonate with vegetables in order to correct metabolic acidosis and delay CKD progression; (4) the reduction of the number and the doses of drugs and chemical substances; and (5) the lowering of urea levels, the cure of intestinal microbioma, and the reduction of cyanates levels (such as indoxyl-sulphate and p-cresol sulphate), which are the most recent known advantages achievable with NT. In conclusion, NT and especially very low protein diet (VLPD) have several beneficial effects in CKD patients and slows the progression of CKD.

## 1. Introduction

Nephrologists replaced the Shakespearean doubt “to be or not to be” with “nutritional therapy (NT) in chronic kidney disease (CKD) patients: yes or not?”

There is still a great debate among nephrologists regarding the efficacy of NT in CKD patients [1,2,3]. Nevertheless, NT has a fundamental role in this medical setting and it has been used for years in the clinical practice with several goals. Initially, NT was basically used with the purpose of administrating a reduced amount of high biological value proteins in order to reduce endogenous urea generation and uremic symptoms [4]. Later, the interest of nephrologists focused on the possibility of delaying CKD progression with NT, but with fluctuating results [3].

As far as we know, there are no publications in the scientific literature in which NT is considered as a pharmacological therapy, a coadjuvant treatment, or a tool to reduce the number and the doses of medications. In fact, a high percentage of CKD patients are treated with diuretics to reduce edema, anti-hypertensive drugs, ACE-inhibitors to treat proteinuria, phosphate binders to prevent secondary hyperparathyroidism and vascular calcification, sodium bicarbonate to correct metabolic acidosis, and erythropoietin to treat anemia [5]. On the other side, there is a growing evidence that serum urea is a toxin [6,7] and there is no pharmacological therapy to treat and reduce its levels. However, NT has several metabolic effects that may lead to the reduction or withdrawal of drugs in some cases. For example, a low-sodium content diet may reduce edema, the amount of diuretics needed and blood pressure levels [8,9]. A low-protein content diet allows phosphate levels correction [10]. A high intake of fruit and vegetables supplies an adequate quantity of bicarbonate to correct metabolic acidosis and to delay CKD progression [11]. Finally, a very low protein diet (VLPD) may allow the reduction of erythropoietin dose [12]. Based upon these observations, the efficacy of NT is evident and recent studies show the advantageous metabolic effects induced by NT and its role in delaying CKD progression [13,14,15]. We will discuss them below point by point.

## 2. Urea Reduction

Figure 1 shows how VLPD is able to reduce urea levels, and in some cases, to normalize them in CKD patients.

Serum urea levels promote cyanate production through the carbamylation of lysine with the formation of homocitrulline (alternative pathway to myeloperoxidase) [7]. High levels of cyanates induce endothelial dysfunction, increase cardiovascular risk, and predispose CKD patients to atherosclerosis and pro-thrombotic alterations [7,16]. As shown in a randomized control trial (RCT), NT allowed a constant reduction of urea levels (up to 61% with the use of a VLPD) with a consensual reduction of cyanates of about 20–30% [17]. The above-mentioned RCT included 60 patients with a mean age of 66 ± 16 years, 46 out of them male and 24 out of them diabetics; patients were randomized with a crossover design to free diet, Mediterranean diet, or VLPD. The study showed that urea levels reduction obtained with VLPD induced a decrease of homocitrulline (marker of protein carbamylation) of 30% compared to Free and Mediterranean diet; the level of urea under which the effect is null was <100 mg/dL (Figure 2) [17]. There were no drugs with the same efficacy or able to induce a reduction of protein carbamylation caused by high urea levels. Moreover, the regression analysis showed that urea reduction correlated with homocitrulline lowering (β-coefficient 4.28; standard error 1.32; *p*-value 0.002).

## 3. Metabolic Acidosis and Insulin Resistance

Several studies evidence the direct role of some uremic toxins, such as urea, trimetylamine N-oxyde, and p-cresol sulphate in glucose homeostasis alterations and incidence of diabetes [18,19]. A recent study showed that NT contributed to the insuline resistance reduction (estimated by the homeostatic model assessment-Homa-test) in 145 CKD patients with type 2 diabetes and a mean age of 65.5 ± 11.4 years. Homa test had a non-linear relation, but after adjustments for confounding factors, bicarbonate levels of 24–28 mmol/L correlated with a significant reduction of Homa test [20]. As a consequence of both serum urea levels reduction and metabolic acidosis correction, our group showed that a vegetarian NT as VLPD allowed a significant reduction (more than 50%) of bicarbonate dose, needed as a drug [21]), being fruit and vegetables rich in bicarbonate, as already previously shown in several papers [22,23].

## 4. Phosphorus and Vascular Calcification

Urinary phosphate excretion in CKD patients is proportional to the residual renal function; therefore, its elimination proportionally reduces and increases serum levels in the absence of a consensual decrease of phosphate dietetic intake. The toxicity of high phosphate levels is well acknowledged [24] and phosphate binders are widely prescribed and used in CKD patients. An Independent Study’s [25] database analysis, including 212 CKD stage 3–4 patients, evidenced 17 patients that were assuming VLPD (10 out of them treated with sevelamer and 7 with calcium carbonate). During the follow-up, the 17 patients reduced sevelamer doses of 39% compared to baseline and calcium carbonate doses of 28%, with serum phosphate levels between 3.5 and 5.5 mg/dL, according to the protocol.

A further positive effect of VLPD was described on the reduction of FGF23 levels [26]. In 32 patients shifted from a low-protein diet (LPD) (0.6 g/kg body weight/day) to a VLPD 0.3 g/kg body weight/day), phosphate levels reduced of 12%, urinary phosphate of 34%, and FGF23 of 33%, the calories intake being equal [26]. The multivariate analysis showed a significant correlation between phosphate levels increase (OR 1.11; 95% CI 1.04–1.19; *p*-value 0.03), urinary phosphate increase OR 1.22; 95% CI 1.12–1.37; *p*-value 0.02), and use of NT with a higher protein content (OR 1.85; 95% CI 1.51–2.23; *p*-value 0.005) (26).

Use of NT plant-derived allowed a less intestinal absorption of phosphate and then a better phosphate levels control with a lesser use of drugs [5].

## 5. Gut and Kidney

Several recent papers focus on the tight connection between gut and kidney, showing that intestinal endothelial alterations and its permeability are caused by both intestinal microbioma modification and high urea levels in CKD patients [6,27,28,29]. Intestinal microbioma is strongly influenced by nutritional habits and by quantity and quality of proteins and fibers assumed with the diet [30].

Our group already showed that VLPD reduced p-cresol sulphate levels of about 30–35% in CKD patients already after one week of dietetic treatment [31].

A recent study confirmed that VLPD significantly reduced indoxyl-sulphate levels of 72% (from 0.46 ± 0.12 to 0.13 ± 0.05 mcg/mL, *p* = 0.002) and p-cresol sulphate levels of 51% (*p* < 0001) compared to Free Diet and Mediterranean diet [32]. Moreover, it was evidenced that this effect was due to urea levels reduction obtained with a NT derived-plant, as VLPD. Similarly, the use of short-chain fatty acids (SCFA) had beneficial effects on intestinal microbioma and reduced the production of inflammatory cytokines in dialysis patients [33]. Acetic acid, butirric acid, propionic acid, and saturated fats with an aliphatic chain of less than 6 carbon atoms are mainly produced through fermentation of dietary fibers and of other non digestible carbohydrates, by intestinal bacteria (particularly, those resistant starch, such as pectin and fructo-oligosaccharides). VLPD supplies vegetables and fruit with a consequent production of a great amount of SCFA that have antinflammatory effects on epithelial intestinal cells; at the same time, VLPD reduces urea levels with an additive beneficial effect on intestinal microbioma [6,34].

## 6. Cardiovascular Risk Reduction

The reduction obtained with the use of NT of several molecules derived by the cellular metabolism, such as urea, trimetylamine N-oxyde, p-cresol sulphate, and cyanates, have a positive impact on inflammatory status and pro-thrombotic events, with the consequent effect of cardiovascular risk reduction in CKD patients [35]. Moreover, also a decreased sodium and phosphate intake, especially if it leads to the control of proteinuria, has an impact on cardiovascular risk reduction [36,37,38,39]. Bellizzi et al. showed that VLPD allowed for a significant reduction of sodium intake (with a reduction of the excreted fraction of sodium of 27%) and blood pressure levels after six months of NT. In fact, they registered an increase of 28% of patients with blood pressure <130/80 mmHg in VLPD group compared to any variation in low-protein diet (LPD) group and a reduction of the number of antihypertensive drugs from 2.6 ± 1.1 to 1.8 ± 1.2 [9].

Moreover, VLPD permitted the reduction of erythropoietin doses of 35% through a better control of phosphate levels and secondary hyperparathyroidism [12].

Finally, Di Iorio et al. showed that the reduction of phosphate intake gained with VLPD led to a reduction of proteinuria (from a median of 1910 to 987 mg/day; *p* < 0.001) in 99 subjects already treated with both ACE-inhibitors and sartans [39]. There were neither changes in GFR nor variation in any other confounding factors and a tight correlation between reduction of urinary phosphate excretion, serum phosphate levels, and proteinuria was found (*p* = 0.04) [39].

## 7. VLPD’s Safety

MDRD study [40] showed that VLPD had no additional benefit compared to a normal protein intake. Furthermore, a MDRD post hoc analysis [41] concluded that VLPD increased the risk of death in CKD patients without a scientific and valid hypothesis of a cause-effect relation between a brief exposure to VLPD and long-term outcomes [42]. Additionally, the long period of time during which patients did not assume a nutritional therapy and several confounding factors (such as absence of information on transplant, dialysis initiation, type of dialysis, vascular access, and others) did not allow a scientific and decisive conclusion on VLPD’s safety. Finally, patients did not show any nutritional deficiency [42]. Later, several studies rejected the conclusions of Menon’s study [43,44,45].

In fact, a historical cohort-controlled study, included patients at dialysis previously treated with VLPD (*n* = 184) or without VLPD (*n* = 334), and unselected patients (control group, *n* = 9092); the principal outcome was survival rate during end-stage renal disease associated to VLPD. Authors showed that VLPD during CKD did not increase mortality in the subsequent dialysis period. The propensity score methods and Cox regression analysis were used to match groups at dialysis start to perform survival analysis and estimate adjusted hazard ratio [44]. Other researchers confirmed Bellizzi’s data [37,44,45].

The possibility of PEW with the use of VLPD is linked only to an incongruous administration of calories. The prescription of VLPD must be accompanied by a correct administration of calories (30–35 kcal/kg × bw/day) [3,29,37,44,45].

We evaluated 90 slow progressors patients (observation period longer than 36 months before initiating death and renal death estimation in LPD and VLPD), referring to the Nephrology Division of Solofra Hospital (Av, Italy); 61 of them were treated with a low-protein diet and 29 of them with a VLPD. Basal data of both groups are shown in Table 1. Figure 3 shows the percentage of death or dialysis start after 36 months of observation. In LPD group, all patients died (*n* = 33) or started dialysis (*n* = 28) within 108 months after observation’s initiation. On the other side, in VLPD group, 5 patients died, 10 patients started dialysis, and 4 patients were still living and not on dialysis at month 132 (*p* < 0.01). We observed a life sparing or dialysis-free time of 24 months/patients with a GFR loss of 1.8 ± 0.7 in LPD group, compared to 1.02 ± 0.9 in VLPD group (*p* < 0.047).

## 8. VLPD Composition

VLPD is a vegetarian therapy composed of vegetables (fresh fruit and vegetables) and the addition of essential aminoacids and ketoanalogues of essential aminoacids. The bromatological composition of VLPD is as follows: energy 30–35 Kcal/kg/day; proteins 0,3 g/kg body weight/day (only of plant origin); lipids 70–80 g/day (only 4% of them saturated); carbohydrates 300–350 g/day fibers 16 g/day; sodium 30–40 mmol/day; potassium 35–50 mmol/day; phosphorus 350–450 mg/day; calcium 200–250 mg/day; iron 3–5 mg/day [9], essential aminoacids, and ketoanalogues mixture 1 pill each 5 kg of body weight. The composition of aminoacids pills is the following: calcium keto-isoleucine 67 mg, calcium keto-leucine 101 mg, calcium keto-alanine 68 mg, calcium keto-valine 86 mg, calcium hydroxyl-methionine 59 mg, L-lysine monoacetate 105 mg, L-threonine 53 mg, L-histidine 38 mg, and L-tyrosine 30 mg. The inclusion of amino acids as oral supplements results in an additional source of nitrogen, with a mean total protein prescription (from food and supplements) of 0.35 g/kg/day in the VLPD group [9].

## 9. Conclusions

In conclusion, NT has several beneficial effects, as above listed; on the other side it may have only two side effects: hyperpotassiemia [46] and/or protein-energy wasting (PEW) [47]. In the absence of anuria due to comorbidities or intercurrent events, hyperpotassemia in CKD is caused by the use of some drugs (alone or in association) such as potassium sparing (ACE inhibitors, anti-aldosteronic diuretics), but not by the simple assumption of vegetables and fruits. At the same time, PEW is caused by an incongruous reduced energy intake [29,37,44,45].

NT is safe if managed by expert nephrologists together with the motivation of the patient [43,47] and only a scarce knowledge of its benefits may discourage to prescribe it [48,49,50].

Finally, if NT is useful to reduce the quantity of pills assumed and the use of drugs in CKD patients (such as antihypertensive drugs, phosphate binders, erythropoietin, and diuretics), it will already be a right motivation to give a positive answer to our initial Hamletic question. A very recent meta-analysis confirms this statement [50].

## Figures and Tables

**Figure 1 jcm-08-00718-f001:**
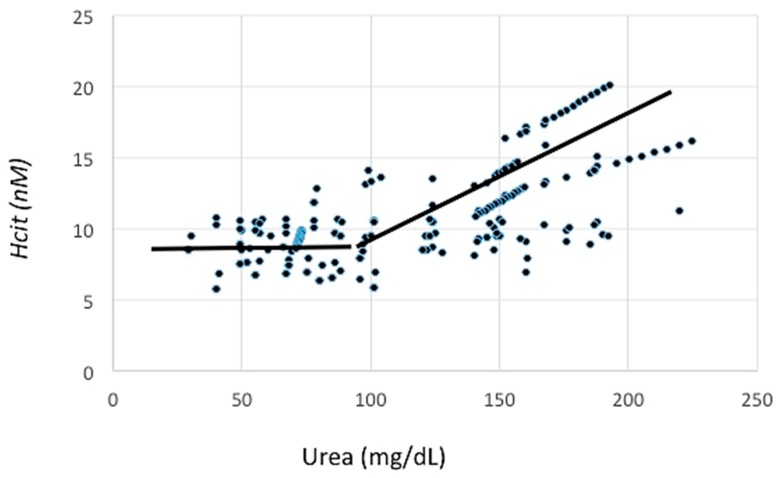
Variation of homocitrulline and urea and break point.

**Figure 2 jcm-08-00718-f002:**
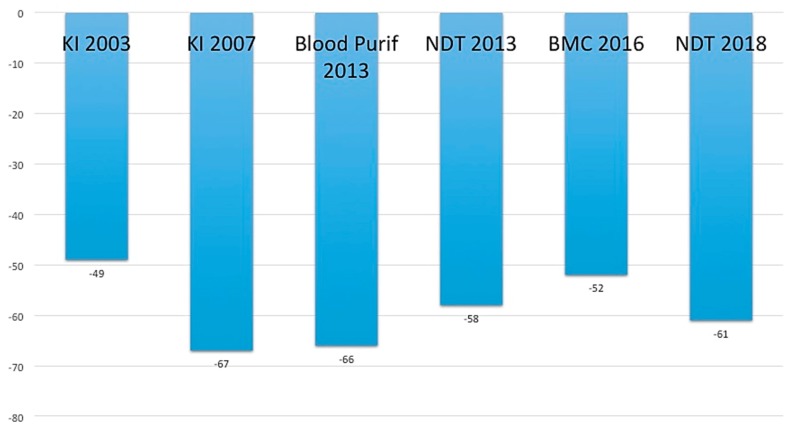
Urea reduction in several published studies. KI 2003 is reference number 12. KI 2007 is reference number 9. Blood Purification 2013 is reference number 31. NDT 2013 is reference number 41. BMC 2016 is reference number 20. NDT 2018 is reference number 17.

**Figure 3 jcm-08-00718-f003:**
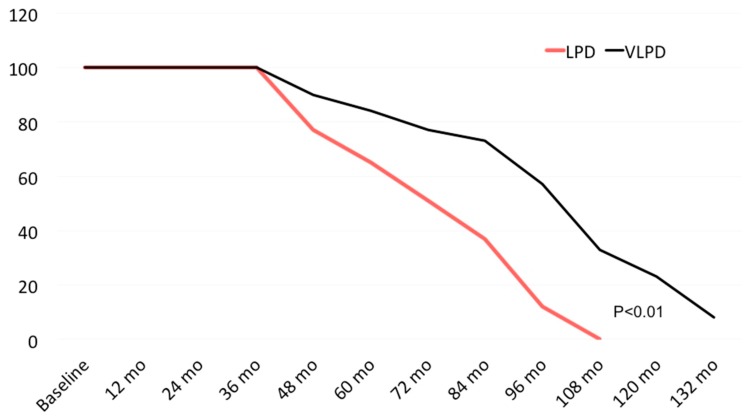
Death or dialysis in 61 patients treated with low-protein diet (0.6 g/kg/day) and 29 of them with Very Low Protein Diet (VLPD) (0.3 g/kg/day).

**Table 1 jcm-08-00718-t001:** Baseline data.

	LPD	VLPD
Number	61	29
Males, *n*	33	16
Age, Years	57 ± 12	55 ± 10
Body Weight, kg	70 ± 7	69 ± 9
Systolic Blood Pressure, mm Hg	137 ± 11	135 ± 16
Diastolic Blood Pressure, mm Hg	78 ± 9	77 ± 11
Diabetes, *n*	14	6
Clearance creatinine, mL/min	34 ± 9	36 ± 12

LPD: low protein diet; VLPD: very low protein diet.

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
