# Peer review of "Very Low Protein Diet for Patients with Chronic Kidney Disease: Recent Insights"

_jcm, 2019, doi:10.3390/jcm8050718_

Round 1
Reviewer 1 Report
Dear Authors,
Following the review of your manuscript entitled: „Very low protein diet for patients with chronic kidney disease: recent insights” I recommend that it should be revised taking into account the following points:
1. Page 1 (Abstract) line 13 and line 26 (Introduction): Why you write in capitals ? Nutritional Therapy (NT) please change it to nutritional therapy (NT)
2. Page 2, line 57 please explain first use of abbreviation RCT (randomized control trial)
3. Page 2 line 61 and page 4 line 112 change Free Diet to free diet
4. Page 2 line 63 change Mediterranean Diet to Mediterranean diet (see line 61)
5. Page 3 line 76 change Homeostatic to homeostatic and Diabetes to diabetes
6. Page 3 line 94 you wrote „LPD (0.6 g/kg body waight/day to a VLPD 0.6 g/kg….; the correct is 0.3 g/kg…)
7. Page 5 line 167 correct the Legend to Figure 3: low protein diet and very low protein diet have the same value of protein (0.3 g/kg/day)
8. Page 6 line 172 and 177 change chetoanalogues to ketoanalogues
9. Please correct the References section according the Journal guidelines and unify all journals data: - give full last pages in Ref. 4, 9, 10-12, 23, etc.; - Correct Ref. 22 and 45 changing volumen, pages and year to year, volumen and pages; - remove editional numbers from selected references (23, 41, 43, 44, 47, etc.); Ref. 37 correct year; Please add dots after last abbreviation of journals – see Ref. 18, 22, 45, 51; Ref. 25 change Neprol. to Nephrol. and Ref. 45 Disease to Dis.; don’t write in capitals titles of papers – see Ref. 8, 14, 19, 21, 30, 33, 35, 36, 38, 39, 50.
10. All references in the text should be identified using numbers in square brackets (e.g. [1-3], [4] etc.)
11. English language shoud be professional corrected, for example:
line 30 is "on the other side", shoud be "on the other hand"
line 46 is "these observation" shoud be "these observations"
line 47 is "it is evident the efficacy of NT" shoud be "the efficacy of NT is evident"
line 75 is "contributed to reduce insuline resistance" shoud be "contributed to the reduction of insuline resistance" or "contributed to the insuline resistance reduction" .
Author Response
My colleagues and I thank the reviewer for his flattering judgment for our paper and, above all, for the kind corrections he suggested.
I made the suggested corrections that are highlighted in red.
Reviewer 2 Report
1. Expect VLPD, if any NT method or clinical treatment can decrease cyanates levels? please discuss this issue
2. Many studies showed that clinical relevance of sarcopenia and anemia in CKD patients. VLPD treatment can progress towards sarcopenia and anemia. Please discuss or analyze the clinical sarcopenia and anemia in CKD patients with VLPD treatment.
3. VLPD treatment can reduce the cyanates levels, however, whether VLPD can improve patients' condition (CKD progression)
4. In line 93-95 and line 167-168 (figure 3), please check the value of LPD and VLPD. In the two parts, the values of LPD and VLPD is the same. Whether mistype the values in LPD or VLPD.
Author Response
My colleagues and I thank the reviewer for his flattering judgment for our paper and, above all, for the kind corrections he suggested.
I made the suggested corrections that are highlighted in red.
For the questions that the reviewer has set, here are our considerations point by point:
1. Expect VLPD, if any NT method or clinical treatment can decrease cyanates levels? please discuss this issue
Answer.
We have shown that the reduction of azotemia allows a reduction of cyanates (derived from the urea-dependent pathway) and that the greater the reduction in urea, the lower the levels of cyanates. In CKD patients, very low values of urea (in some patients even normal serum urea values are reached) can be obtained only with a very low protein intake, as happens in VLPD.
In the experiment (reference n °17) we used the comparison between Mediterranean diet and VLPD just to exclude the intervention of dietary fibers (which is similar in the two nutritional therapies)
2. Many studies showed that clinical relevance of sarcopenia and anemia in CKD patients. VLPD treatment can progress towards sarcopenia and anemia. Please discuss or analyze the clinical sarcopenia and anemia in CKD patients with VLPD treatment.
Answer.
The hypoproteic nutritional therapy, not even VLPD, is not in itself a cause of PWE and / or anemia if a correct quantity of calories is administered (30-35 cal / kg bw / day).
In fact, in one of our previous papers (reference 12) we showed that the duration of follow-up was significantly longer in VLPD than LPD (23.2 months vs. 19.6 months, respectively, P = 0.02); in VLPD, in fact, eight out ten patients completed the 24-month period of fol- low-up, while seven out ten patients of the LPD group stopped the study after month 18 having reached the end point.
3. VLPD treatment can reduce the cyanates levels, however, whether VLPD can improve patients' condition (CKD progression)
Answer.
Urea participates directly in the pathogenesis of cardiovascular alterations in CKD subjects through the reaction between cyanates and ammonium which leads to the formation of isocyanic acid with protein carbamylation, substances of high atherosclerotic impact. Therefore the production of Cyanates certainly reduces patients' condition (and increase the speed of CKD progression)
Reviewer 3 ReportThe manuscript written by Micco et al in the use of nutritional therapy (especially VLPD) in CKD patients is in excellent quality. It’s a well written review and includes sufficient references. Few minor suggestions 1) the abstract lacks the conclusion of the manuscript, its basically includes the goals and advantage, but not the conclusion 2) line 171-182 is not a conclusion, so it should be placed in a separate title possibly ‘VLPD dietary composition or similar title’.
Author Response

(The authors gave the same response as above.)
